

# Gait recognition using a few gait frames

Lingxiang Yao[1], Worapan Kusakunniran[2], Qiang Wu[1] and Jian Zhang[1]

[1] School of Electrical and Data Engineering, University of Technology Sydney, Sydney, Australia
[2] Faculty of Information and Communication Technology, Mahidol University, Nakhon Pathom, Thailand

## ABSTRACT

Gait has been deemed as an alternative biometric in video-based surveillance applications, since it can be used to recognize individuals from a far distance without their interaction and cooperation. Recently, many gait recognition methods have been proposed, aiming at reducing the influence caused by exterior factors. However, most of these methods are developed based on sufficient input gait frames, and their recognition performance will sharply decrease if the frame number drops. In the real-world scenario, it is impossible to always obtain a sufficient number of gait frames for each subject due to many reasons, e.g., occlusion and illumination. Therefore, it is necessary to improve the gait recognition performance when the available gait frames are limited. This paper starts with three different strategies, aiming at producing more input frames and eliminating the generalization error cause by insufficient input data. Meanwhile, a two-branch network is also proposed in this paper to formulate robust gait representations from the original and new generated input gait frames. According to our experiments, under the limited gait frames being used, it was verified that the proposed method can achieve a reliable performance for gait recognition.

# INTRODUCTION

Gait has many significant advantages over other forms of biometric. First, gait can be acquired at a distance in a non-invasive manner without subject cooperation. It is easy to be acquired but hard to be distinguished as walking is a common but unique activity of each subject. More importantly, gait still works well even if it is obscured and/or its resolution is low. As an example, in Denmark and UK, gait analysis has been used to collect evidences of convicting criminals (*Larsen, Simonsen & Lynnerup, 2008*; *Bouchrika et al, 2011*).

For years, many traditional gait recognition approaches have been published. Based on the gait features generated, these approaches can be classified into two categories, including appearance-based methods and model-based methods.

In the first category, appearance-based methods demand an image pre-processing of segmenting human silhouettes from gait images or videos. Gait Energy Image (GEI), the most widely-used gait feature in this category, is an average of consecutive silhouettes within a complete gait period(s) (*Han & Bhanu, 2005*). Motion Silhouette Image (MSI) (*Lam & Lee, 2006*) is another similar gait feature, in which each pixel is denoted as a function of motion in the temporal dimension over all silhouettes within an entire gait period. An advantage of this category is that these features will offer a large number of discriminative

Corresponding author
Worapan Kusakunniran,
worapan.kun@mahidol.edu

information for gait recognition, which contributes to a robust recognition performance under controlled environments without any changes of external factors. However, in the real world, due to occlusion and illumination, it is difficult to get perfect/complete human silhouettes, and such low-quality silhouettes surely will have a bad influence on gait recognition.

In the second category, model-based methods require a pre-defined human model/pattern which reveals human body's structures and linkages between different body parts. Handling human legs as a dynamically-interlinked pendulum model (*Cunado, Nixon & Carter, 1997*), gait features are developed from frequency variations of a thigh inclination with phase-weighted Fourier magnitude spectrum. In *Wang et al. (2004)*, the static body information is processed by Procrustes Shape Analysis (PSA), and the dynamic information is generated by tracking each subject and recovering the joint-angle trajectories of their lower limbs. Moreover, there exists a great potential in reconstructing 3D human models (*Urtasun & Fua, 2004*; *Zhao et al, 2006*). The major disadvantage of this category lies in that there remains a huge challenge in extracting the underlying models from gait images and videos (*Yam & Nixon, 2009*; *Wang et al, 2012*). The predicted locations for human joints are not always robust based on markerless motions (*Yam & Nixon, 2009*). In addition, model-based methods always perform worse than appearance-based methods in many gait datasets. One reason is that in most model-based methods only sparse keypoint information is applied. Another reason is that in these gait datasets there exists a little difference in each person's silhouettes, and thus human appearance can be seen as the most discriminative gait feature representations (*Lombardi et al., 2013*).

Recently, deep learning-based approaches are becoming flourishing in the computer vision community. Generally, these approaches are advantageous to gait recognition in two manners. First, they are conductive to supplying high-quality human silhouettes and skeletons for gait recognition, e.g., (*He et al., 2017*; *Gong et al., 2017*; *Cao et al., 2017*; *Fang et al., 2017*). Second, many CNN-based methods also have achieved an excellent performance in gait recognition. For example, *Shiraga et al. (2016)* proposed GEINet using GEI as input. The method in *Wolf, Babaee & Rigoll (2016)* designs a three-dimensional network using special inputs, which fuse the gray-scale images and optical flows in the horizontal and vertical directions. The method in *Castro et al. (2017)* proposes a network with the spatio-temporal cuboids of optical flows as the input. All these proposed networks have presented a better performance than the aforementioned traditional methods in many gait recognition benchmarks.

Lately, GaitSet proposed in *Chao et al. (2019)* has shown the state-of-the-art result for gait recognition. A new perspective is illustrated in *Chao et al. (2019)*, i.e., a gait can be taken as a set of independent frames. Based on this set perspective, GaitSet attempts to learn identity information from each input set. Because of the set attributes, GaitSet is immune to permutation of frames, and can integrate frames collected in various walking conditions. However, as claimed in *Chao et al. (2019)*, GaitSet cannot work well if the input frame number is small. Figure 1 presents the relationship between the number of silhouettes in each input set and the corresponding averaged rank-1 accuracy of all probe views. The accuracy first monotonically raises as the number of silhouettes increases. It reaches a

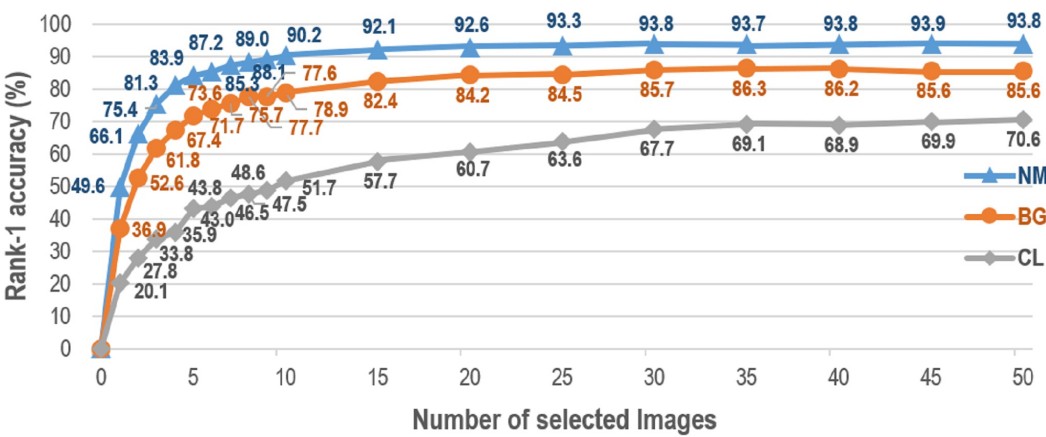

**Figure 1** **Average rank-1 accuracies with constraints of silhouette volume on CASIA-B in the setting LT.** Accuracies are averaged from all views excluding identical-view cases across 10-times experiments.

dropped accuracy of 87.2% when taking a small number of only seven silhouettes as the input. For GaitSet, each input silhouette is independently processed in the shallow layers. In order to aggregate the features of each silhouette into a feature of the whole input sequence, a set pooling function is specially utilized to select the most discriminative parts from each processed silhouette. As more input silhouettes can be utilized, a more complete description can be generated for each sequence, and the discrimination capability of GaitSet can be gradually increased. On the other hand, for the cases of only a few silhouettes being utilized, the discrimination capability of GaitSet can be remarkably decreased. This is because these silhouettes can only occupy a small proportion of each input whole sequence, and the aggregated features cannot establish a comprehensive description for each sequence. Hence, it has become an urgent problem for gait recognition how to achieve a robust performance when only a few input frames can be offered.

This paper focuses on how to improve the gait recognition performance when only a few gait frames can be obtained. It is important to enhance the performance for gait recognition when the number of gait frames available is limited. From a practical perspective, due to problems of occlusion and illumination, it is highly impossible to obtain enough high-quality frames for an effective gait recognition. Also, the frames we can obtain are limited by individuals' optional walking speeds and cameras' different frame rates. On the other hand, technically speaking, it is difficult to generate a perfect human silhouette and skeleton model for gait recognition, although lately lots of deep learning-based methods have been proposed to approach these two problems. Segmentation methods may fail due to the diversity of dressing patterns, and skeleton extraction might not work well due to the existence of self-occlusion. Therefore, it is necessary to propose an efficient gait recognition method with only a few gait frames being used.

In recent decades, many methods have been proposed for gait recognition to address the problem of low frame rates. For example, in *Makihara, Mori & Yagi (2010)*, *Akae, Makihara & Yagi (2011)*, *Akae et al. (2012)*, *Xu et al. (2020)*, the problem of low frame rates is handled

using temporal reconstruction and super-resolution methods. In *Guan, Li & Choudhury (2013)*, gait features are directly formulated from the input low frame-rate sequences via metric learning methods. In *Babaee, Li & Rigoll (2019)*, a CNN-based network is proposed to progressively reconstruct a complete GEI of a full gait cycle from an incomplete GEI of a low frame-rate gait sequence. In this paper, however, we give more attention to the problem of insufficient input data. As stated above, low frame rates can be one of the reasons that cause this problem. Considering the development of cameras and transmission bandwidth, the effect of low frame rates on this problem is becoming diminishing these days. Other factors, e.g., failure of segmentation and skeleton extraction algorithms, are becoming increasingly influential on this problem.

Motivated by *Makihara, Mori & Yagi (2010)*, *Guan, Li & Choudhury (2013)*, a robust gait recognition method is proposed in this paper with limited input frame number. In contrast to *Makihara, Mori & Yagi (2010)* reconstructing a cycle of a high frame-rate sequence through temporal super-resolution methods, in our method we directly generate more input frames without any computation of gait cycles. In contrast to *Guan, Li & Choudhury (2013)* enhancing the feature robustness using metric learning methods, in this paper the most discriminative features are directly formulated from the original and generated gait frames through a two-branch network. Specifically, frame-level features are first independently extracted from each input frame. A max-pooling function is applied to integrate these extracted frame-level features into a set-level feature of each branch. Meanwhile, Horizontal Pyramid Mapping (HPM) is also used in our method to map the set-level features into a more discriminative subspace. The final feature descriptor is formed by concatenating the features developed from each branch. Besides, three different strategies are selected and compared in the process of more frames being generated. Two typical image augmentation methods, horizontally flipping silhouettes and generating more silhouettes using GAN-based networks, have been introduced into our paper to increase the data volume and overcome the problem of insufficient input data. Also, motivated by the aforementioned appearance-based and model-based methods, a special input is utilized in this paper for gait recognition by assembling segmented human silhouettes and extracted skeleton models. It is reasonable to assemble these two inputs for gait recognition, since human silhouettes can offer sufficient discriminative spatial features, and skeleton models will supply robust knowledge of human body structures. Compared with other methods handling this insufficient frame problem without including such skeleton information, a more complete and comprehensive gait description can be established for each input sequence by combining these spatial and body structure features. Relevant experiments also verify that compared with the other two strategies, it is more functional to combine human silhouettes and their skeleton models as the network input when the accessible frame number is restricted to other exterior factors.

The main contributions of this paper are summarized as follows.

- This paper further explores gait recognition with insufficient input frames. It is significant to improve the gait recognition performance when only a few gait frames are

available, because in the real world, due to many reasons, it is unlikely to always obtain sufficient gait frames for each person.

- In this paper, a robust gait recognition method is developed to tackle the problem of insufficient input data. First, three strategies are proposed to create more input frames and eliminate the generalization error caused by insufficient input data. A two-branch network is also proposed in this paper to extract robust features from the original and new generated gait frames.

- Experiments on the CASIA Gait Dataset B and the OU-MVLP Dataset proves that under limited gait frames being used, the proposed method can achieve a reliable performance for gait recognition.

## RELATED WORK

### Low frame-rate gait recognition

In decades, many methods have been proposed to tackle the problem of low frame rates for gait recognition. Basically, these methods can be divided into three different categories, including temporal interpolation and super-resolution-based methods, metric learning-based methods, and directly gait feature reconstruction-based methods (*Xu et al., 2020*).

For methods in the first category, temporal reconstruction and super-resolution techniques are imported to address the problem of low frame rates. In *Makihara, Mori & Yagi (2010)*, a temporal super-resolution method is proposed from quasiperiodic image sequences. Based on phase registration, low frame-rate sequences from multiple periods is exploited to integrate one period of a high frame-rate sequence via energy minimization. In *Akae, Makihara & Yagi (2011)*, an exemplar of a gait image sequence with a high frame-rate is introduced to suppress the wagon wheel effect caused by extremely low frame-rate cases. Besides, in *Akae et al. (2012)*, a unified framework of example-based and reconstruction-based periodic temporal super-resolution is established to further reduce the stroboscopic effect. However, these proposed methods can only guarantee the optimality of reconstruction quality rather than recognition accuracy, which is the most important for gait recognition. In order to ensure the reconstruction quality and recognition accuracy simultaneously, a unified framework of a phase-aware gait cycle reconstruction network (PA-GCRNet) is created in *Xu et al. (2020)*. Taking the phase of a single input silhouette into account, a full gait cycle of a silhouette sequence with corresponding phases is first reconstructed by PA-GCRNet. After that, all reconstructed silhouettes are input into another recognition network to learn more discriminative features for gait recognition.

For methods in the second category, gait videos of low frame-rates are directly approached using metric learning techniques. In *Guan, Li & Choudhury (2013)*, a single gray-scale image is first extracted as the input template, and a random subspace method (RSM) is introduced to reduce the generalization error caused by low frame rates. The rationale behind is that the error caused by low frame rates can be deemed as a kind of intra-class variations, which can be processed by a general model, e.g., RSM in *Guan, Li & Choudhury (2013)*.

Methods in the third category can be regarded as a combination of the aforementioned two categories. A feature template is first generated from the input low frame-rate sequences. Based on the feature template, a more effective gait representation can be reconstructed using some reconstruction techniques. For example, by using ITCNet proposed in *Babaee, Li & Rigoll (2019)*, a complete GEI of a full gait cycle will be progressively reconstructed from an incomplete GEI of a low frame-rate gait sequence. However, as we stated above, this method merely focuses on optimizing the reconstruction performance of GEI, thus its recognition accuracy cannot be guaranteed.

In this paper, we pay more attention to the problem of insufficient input data. Videos of low frame-rates can be seen as a manifestation of this insufficient data problem. However, with the development of camera technologies and transmission bandwidth, videos of high frame-rates are becoming increasingly frequent in our daily lives and lots of surveillance systems. The influence caused by low frame rates on gait recognition nowadays is gradually decreasing. Other factors, e.g., performance of segmentation and skeleton extraction algorithms, are becoming more and more influential on this insufficient data problem.

Still, lots of successful practices handling the problem of low frame rates can be introduced to tackle the problem of insufficient input data, especially some temporal reconstruction and super-resolution techniques proposed in the first category. Motivated by these rationales, in this paper, more gait frames are generated to reduce the generalization error caused by insufficient input. But different from the first category generating more frames based on gait cycles, in our method more input frames are generated without any knowledge of gait cycles. Also, different from the second category increasing the feature robustness using metric learning methods, in our method gait features are directly extracted from the original and generated input frames.

## Deep learning-based gait recognition

Recently, deep learning-based methods are becoming flourishing in the computer vision community (*Huang et al., 2018*; *Huang et al, 2019*). Lots of deep learning-based networks also have been constructed for gait recognition to enhance its discrimination ability, and most of these networks have achieved a competitive performance for gait recognition. Based on their input information types, these networks can be divided into three different categories, i.e., averaged templates, optical flows and silhouette sequences.

For networks in the first category, averaged templates are used as their network input. GEI, the averaged template of silhouettes from an entire gait period, has been widely used in many different networks (*Shiraga et al., 2016*, *Zhang et al, 2016*; *Wu et al, 2016*; Zhang et al., 2019). SGEI, the averaged skeleton template from an entire gait period, also has been adopted in some networks (*Yao et al., 2018*; *Yao et al, 2019*). However, these inputs cannot be used for the cases of insufficient gait data, since these inputs are generated from successive gait frames, which cannot be realized in these insufficient input cases.

For networks in the second category, optical flows are utilized as the input (*Wolf, Babaee & Rigoll, 2016*). Similar to averaged templates in the first category, it is not advisable to utilize optical flows to approach the problem of insufficient gait data. Optical flows enable to capture the movements across two or three successive frames. However, when the input

frame number is limited and only intermittent gait frames can be acquired, it gets less likely to figure out the optical flows required by this category.

For networks in the third category, silhouette sequences are directly taken as their input. Each silhouette is first independently processed. A pooling operation is used to integrate the features of each silhouette into a feature of the entire silhouette sequence (*Wu, Huang & Wang, 2015*; *Chao et al., 2019*). Based on the assumption that the appearance of each silhouette has illustrated its position information (*Chao et al., 2019*), these networks are immune to permutation of frames. Considering the discontinuity of insufficient data, it is recommended to directly use the available gait frames as our network input.

## PROPOSED METHODS

As Fig. 1 illustrates, the number of input images has a significant influence on gait recognition performance, especially when only a small number of gait images can be obtained. It is well-founded to identify different persons by gait, because each person exhibits his/her walking pattern in a repeatable and sufficiently unique manner (*Winter, 2009*). The natural walking of a person is periodic (*Babaee, Li & Rigoll, 2019*). However, in a full gait cycle, the walking pattern of a person will gradually change as time goes by. Thus, for gait recognition, insufficient input will increase the generalization error for each person, especially when the available frame number is less than a full gait cycle. Inspired by *Makihara, Mori & Yagi (2010)*, in this paper, more input frames are derived from the original input to reduce this generalization error. Three different strategies are utilized and compared in this paper to generate the most efficient input frames for gait recognition. Meanwhile, inspired by *Guan, Li & Choudhury (2013)*; *Chao et al. (2019)*, a two-branch network is also proposed in this paper to generate the most discriminative gait features from the original and our generated input frames.

### Generating more input frames using different strategies

Data augmentation, which contains a variety of techniques enhancing the size and quality of datasets, offers a data-space solution to the problem of limited data (*Shorten & Khoshgoftaar, 2019*). Generally, based on the used technologies, these techniques can be roughly classified into two categories, including basic image manipulations and deep learning-based approaches (*Shorten & Khoshgoftaar, 2019*). Also, a comparative study is organized In *Shijie et al. (2017)* to show the performance differences of these different techniques. The comparison result shows that under the same conditions cropping, flipping, rotation, and WGAN work much better than the other augmentation techniques.

Data augmentation also has been used in gait recognition to enhance its efficiency. For example, simple image transformations, e.g., horizontal flipping and slight rotation, have been used in *Shiraga et al. (2016)*; *Yao et al. (2018)* to avoid the problem of over-fitting. Also, In *Zhang, Wu & Xu (2019)*, Zhang et al. (2019) GAN-based networks are proposed to increase the generalization strength and eliminate the influence caused by exterior changes. In this paper, two different augmentation techniques are selected to approach the insufficient input problem for gait recognition. First, as one of the most widely-used basic image manipulations, horizontal flipping is easy to implement but has proved efficient on

many different datasets. Also, as the most representative deep learning-based augmentation method, CycleGAN enables to translate an image from the source space to the target space in the absence of paired examples (*Zhu et al., 2017*). It has been utilized in a variety of practical applications, and has presented a remarkable performance in these applications. It is worth mentioning that in this paper, horizontal flipping and CycleGAN are introduced as two strategies to offer more input frames for gait recognition when insufficient input data can be obtained, which is quite different from their original notion of data augmentation. Data augmentation refers to enhancing the diversity of a training set by using certain augmentation method. However, in this paper, horizontal flipping and CycleGAN are applied much similar to the aforementioned temporal reconstruction and super-resolution methods, which generate more input frames from a limited number of input data regardless of training or testing phase. In this paper, since appearance-based methods always present a better result than model-based methods, human silhouettes are first segmented from the original input data. Then more input silhouettes are subsequently generated based on the segmented human silhouettes using horizontal flipping or a retrained CycleGAN model.

Moreover, by combining segmented human silhouettes and extracted skeleton models, a special input is generated in this paper for gait recognition to address the problem of insufficient input data. It is reasonable to assemble segmented human silhouettes and extracted skeleton models for gait recognition, because these two inputs depict gaits from two different perspectives, and a complete description can be provided for each person through this input combination. As mentioned above, sufficient discriminative body shape hints can be offered by human silhouettes. But since human silhouettes are deeply entangled with appearances, these body shape hints are deeply affected by external clothing changes. Meanwhile, robust knowledge of human body structures can be offered by extracted skeleton models. Although the human body knowledge exhibits the robustness to appearance changes, it still suffers from the problem of sparsity. Considering the strengths and weaknesses of each input, a tendency of mutual utilization and promotion can be formed between these two inputs by combining them together. In this way, a full-scale gait description can be formulated for each person. Specifically, human silhouettes can be directly segmented from the original input data using (*Gong et al., 2017*; *He et al., 2017*). Meanwhile, based on human anatomy and biomechanics, skeleton models are established in this paper by linking the correlative skeleton keypoints using ellipses. Methods such as (*Cao et al., 2017*; *Fang et al., 2017*) can be adopted to generate our required skeleton keypoints. In this paper, 17 skeleton keypoints are first extracted. Then, each limb is connected using an ellipse, and its transparency is dependent on the averaged confidence of the connected two keypoints. A lighter colour represents a higher confidence, and correspondingly a darker colour represents a lower confidence.

Figure 2 shows some samples for the same person using the aforementioned three strategies in different walking conditions. Compared with the other two strategies, horizontal flipping enables to offer body shape hints from another different viewing angle, which might be helpful in view-changing gait recognition. Also, without losing most important body shape hints, more silhouettes are successfully generated by CycleGAN from the original segmented human silhouettes. However, since these two strategies are

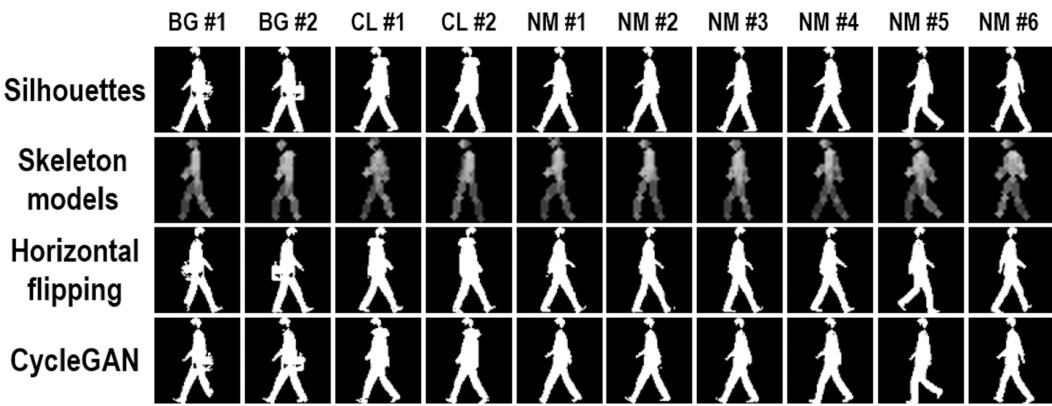

**Figure 2** **Three sample augmentation methods.** The first row shows origin silhouettes from three walking conditions including BG (carrying a bag), CL (wearing a coat), and NM (normal walking). The final three rows show sample silhouettes from three different types of augmentations including skeleton modeling, horizontal flipping, and cycleGAN.

proposed based on human silhouettes, the new generated silhouettes may face the same problem as the original ones. As Fig. 2 indicates, the original segmented and the new generated silhouettes are both prominently influenced by the external clothing changes, which will further influence the performance of gait recognition. Different from horizontal flipping and CycleGAN, skeleton models introduce a new form of gait information to address the problem of insufficient input data, i.e., robust knowledge of human body structures. There are two strengths in combining these skeleton models to approach this insufficient input problem. First, different from human silhouettes, human body structures are more insensitive to external clothing changes. Second, the extracted skeleton models are mainly influenced by the performance of skeleton extraction algorithms, while the new generated silhouettes are influenced by the performance of segmentation algorithms and horizontal flipping (or CycleGAN) at the same time, which may cause error propagation. Therefore, compared with generating more silhouettes through horizontal flipping and CycleGAN, it is more efficient to approach the insufficient data problem by combining segmented human silhouettes and extracted skeleton models.

## Generating gait features using proposed network

In this paper, a two-branch network is also proposed to further handle the problem of insufficient input data. Figure 3 presents the structure of this proposed network. The first branch extracts features from the original segmented human silhouettes, and the second branch extracts features from the new input frames generated using the aforementioned three strategies. It is reasonable to propose a two-branch network to handle these two inputs separately, because errors may occur when the new input frames are being generated, especially for CycleGAN and extracted skeleton models.

As Fig. 3 indicates, due to the discontinuity and disorderliness of insufficient data, it is recommended to extract features from the input frames for each branch directly. Given that GaitSet has achieved a prominent recognition performance and proves immune to

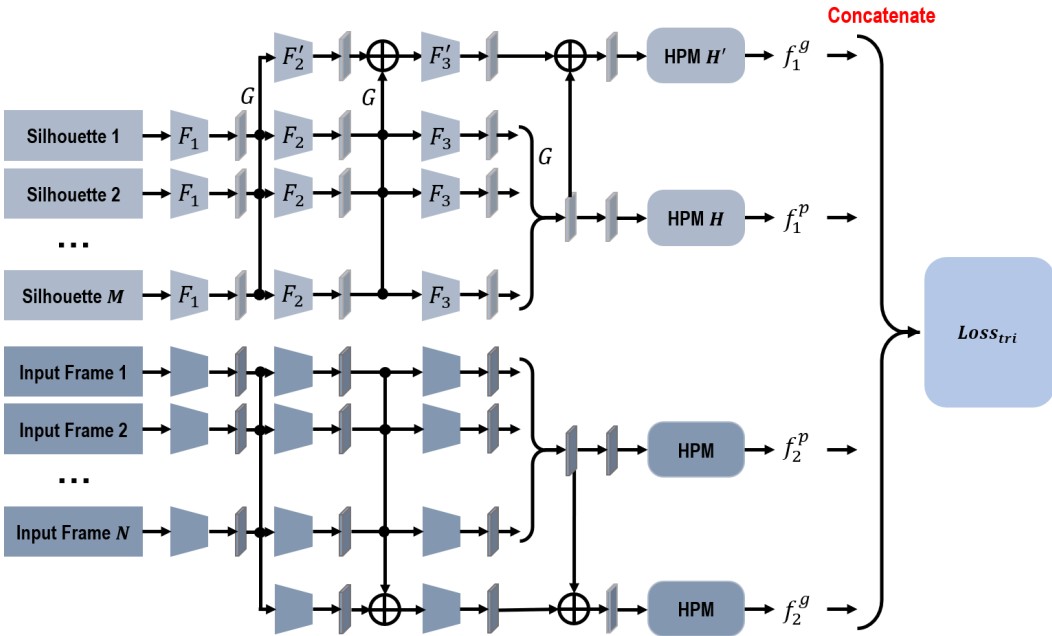

**Figure 3** The network structure of the proposed network.

permutation of frames at the same time (*Chao et al., 2019*), in our network, each branch has a similar structure as GaitSet. Specifically, within each branch, frame-level features are first independently extracted from each input frame. Then, a max-pooling function is utilized to transform the frame-level features into a set-level feature. Meanwhile, a global pipeline is utilized to collect set-level features from different convolution stages. HPM is also introduced to project the set-level features into a discriminative multi-scale subspace. Within each branch, weights are only shared in the procedure of generating frame-level features. The output gait representation is integrated by concatenating four different feature components together. In the training phase, a triplet loss is used to recognize the gait representations of different persons in different walking conditions.

Based on the assumption in *Chao et al. (2019)*, the segmented silhouettes of each person can be tackled as a set $\chi_1 = \{x_i | i = 1, 2, 3, \ldots, m\}$. Correspondingly, the new generated input frames also can be handled as a set $\chi_2 = \{x_i | i = 1, 2, 3, \ldots, n\}$. $m$ and $n$ indicate the cardinality of each input set. Thus, for the first branch, features generated from the first silhouette input set can be formulated as below,

$$f_1^p = H(G(F_3(F_2(F_1(\chi_1))))) \tag{1}$$

$$f_1^g = H'\left(F_3'\left(F_2'\left(G(F_1(\chi_1))\right) \oplus G(F_2(F_1(\chi_1)))\right) \oplus G(F_3(F_2(F_1(\chi_1))))\right) \tag{2}$$

where $F_1$, $F_2$, $F_3$, $F_2'$, and $F_3'$ represent different convolution stages. Among them, $F_1$, $F_2$, and $F_2'$ are made up of two successive convolution layers with another pooling layer, while

$F_3$ and $F_3'$ are simply made up of two continuous convolution layers. $G$ denotes the pooling function of transforming the frame-level features into a set-level feature. $H$ denotes the HPM module of mapping the set-level features into a more discriminative subspace. Moreover, $\oplus$ denotes the pixel-wise adding operation.

As indicated in Fig. 3, the two branches of our proposed network have a similar structure. Thus, features extracted from the second branch, $f_2^p$ and $f_2^g$, can be calculated in a similar way to the first branch. The final output gait representation, $f$, thus can be formulated as below,

$$f = f_1^p \odot f_1^g \odot f_2^p \odot f_2^g \qquad (3)$$

where $\odot$ denotes the concatenation operation.

Compared with GaitSet in *Chao et al. (2019)*, our proposed two-branch network is more efficient when addressing the problem of insufficient input data. For the first branch, discriminative features are integrated from the original segmented human silhouettes. In the meantime, for the second branch, additional features are developed from our new generated input frames. Since only insufficient input data can be used, features generated by the first branch suffer from the influence of inadequate generalization capability. In this paper, more input frames are generated for the second branch, and features developed by the second branch can be used as feature complement to help eliminate this negative influence. For example, features generated from the horizontally-flipping silhouettes can offer more human body shape hints from another different viewing angle. Using these features as complement not only enhances the generalization strength, but also increases its robustness to viewing variances. Besides, features generated from the input skeleton models can provide more robust knowledge about human body structures. It is recommended to use these generated features as complement to enhance the generalization strength of the first branch, since another different form of input information is introduced into the second branch. In addition, because the introduced knowledge of human body structures is more insensitive to appearance variations, the robustness of our final output features also can be improved at the same time.

## EXPERIMENTS

In this section, the aforementioned three strategies and the proposed two-branch network will be verified on two of the most widely-used gait databases, the CASIA Gait Dataset B (*Yu, Tan & Tan, 2006*) and the OU-MVLP Dataset (*Takemura et al., 2018*). These experiments will illustrate that compared with GaitSet, the proposed method is more robust and efficient when handling the problem of insufficient input data.

### Dataset

As one of the most widely-used gait datasets, the CASIA Gait Dataset B captures 13,640 gait videos of 124 persons from 11 viewing angles ($0°$, $18°$, $36°$, …, $180°$). Under each viewing angle, 10 videos are captured for each person, i.e., six videos of normal wearing styles (NM), two videos of wearing a long coat (CL), and two videos of carrying a bag (BG). In addition, the CASIA Gait Dataset B also offers the segmented human silhouettes of each video frame.

Figure 2 indicates some sampled silhouettes from this dataset for the same person under the viewing angle of 90°. In this paper, in order to present a comprehensive comparison of the proposed method and GaitSet, we follow the same LT partition of training and testing sets as (*Chao et al., 2019*). Videos of the first 74 persons are used as the training set, and videos of the rest 50 persons are used as the testing set. For the testing set, the first 4 NM videos of each person are treated as the gallery set, and the rest videos of each person are then regarded as the probe set.

As the world's largest public gait dataset, the OU-MVLP Dataset offers segmented silhouette sequences of 10,307 subjects from 14 viewing angles (0, 15, 30, 45, 60, 75, 90, 180, 195, 210, 225, 240, 255, 270). Sequences are divided into training and testing sets based on subject IDs, 5153 for training and 5154 for testing. For the testing set, sequences with index #01 are used as the gallery set, and sequences with index #00 are handled as the probe set. Also, for the OU-MVLP Dataset, skeleton locations can be directly obtained from (*An et al., 2020*).

## Experiments on the CASIA gait dataset B

In this part, the efficiency of our proposed method is evaluated on the CASIA Gait Dataset B. Our empirical experiments contain two parts. First, we evaluate the performance changes if the same number of inputs are fed into the two-branch network, i.e., $m = n$. Next, we compare the performance when the two branches are fed with different numbers of frames but the sum number is set to 30, i.e., $m + n = 30$. The aforementioned three different strategies are compared in both experiments.

### *Training and testing details*

In order to generate more input silhouettes using CycleGAN, a new model is first retrained by following the same configuration as *Zhu et al. (2017)*. The training set is made up of silhouettes from the first 74 persons. Specifically, the source space is built by all current frames, and the target space is formed by their sequential frames. There is no distinct margin between the source and target spaces, thus more reliable silhouettes can be generated in this paper to approach the problem of insufficient input data.

For skeleton models, humans of each frame are first located based on the provided human silhouettes. A pre-trained AlphaPose model (*Fang et al., 2017*) is directly used to extract the skeleton keypoints from each located human area. Based on the extracted skeleton keypoints, in this paper, skeleton models are generated by connecting the correlative keypoints with ellipses. It is suitable to directly extract the skeleton keypoints using the pre-trained AlphaPose model, because AlphaPose has already presented a notable performance in a more complicated outdoor environment for many types of activities, including a walking activity.

Moreover, for the proposed two-branch network, the input frames are first aligned, cropped, and resized to $64 \times 44$ pixels. In each branch, the channels of each convolution stage are set to 32, 64, and 128. For the two global pipelines, the channels are correspondingly set to 64 and 128. The HPM scales are set to 5.

In the training phase, each time a batch with size of $8 \times 16$ sets is randomly picked from the training set, where 8 means the number of persons and 16 denotes the number of sets

**Table 1** Averaged rank-1 accuracies (%) on CASIA-B, excluding identical-view cases.

| The number of original input frames ($m$) | | 5 | 10[a] | 15 | 20[b] | 25 | 30[c] |
|---|---|---|---|---|---|---|---|
| | Skeleton models | 95.2 | 96.1 | 96.8 | 96.5 | 96.7 | 96.8 |
| NM | Horizontal flipping | 94.3 | 94.8 | 94.9 | 95.9 | 96.1 | 95.9 |
| | CycleGAN | 94.1 | 94.9 | 95.6 | 95.7 | 95.9 | 95.5 |
| | Skeleton models | 88.6 | 90.2 | 90.4 | 90.4 | 91.0 | 91.2 |
| BG | Horizontal flipping | 85.7 | 88.2 | 88.5 | 89.8 | 88.8 | 89.71 |
| | CycleGAN | 86.3 | 88.1 | 88.8 | 88.7 | 89.1 | 89.1 |
| | Skeleton models | 64.8 | 71.2 | 73.5 | 73.6 | 76.1 | 77.2 |
| CL | Horizontal flipping | 57.2 | 64.7 | 69.3 | 72.5 | 74.0 | 75.0 |
| | CycleGAN | 57.3 | 66.0 | 69.9 | 70.8 | 73.2 | 74.9 |

**Notes.**

[a] The accuracies of GaitSet feeding 10 frames are 94.0%, 86.9%, and 62.7% for NM, BG, and CL respectively.

[b] The accuracies of GaitSet feeding 20 frames are 94.7%, 87.9%, and 68.3% for NM, BG, and CL respectively.

[c] The accuracies of GaitSet feeding 30 frames are 95.0%, 87.2%, and 70.4% for NM, BG, and CL respectively.

each person has in this batch. Each set consists of $m$ segmented silhouettes and $n$ generated new input frames. Adam (*Kingma & Ba, 2014*) is chosen as our optimizer, and the learning rate is set to $1e-4$. Meanwhile, the margin of $BA_+$ triplet loss is set to be 0.2 in our training process (*Hermans, Beyer & Leibe, 2017*).

In the testing stage, the batch size is set to 1. Besides, rather than use randomly-sampled silhouettes and new generated input frames, all input data is used to formulate the final output representation.

### Results on the CASIA gait dataset B

Table 1 shows the comparison results when the same number of inputs are fed into the proposed two-branch network. The comparison results indicate that compared with the other two strategies, it is more effective to assemble human silhouettes and skeleton models as input when addressing the problem of insufficient input data for gait recognition, especially in the CL cases. As stated above, the other two strategies are dependent on human silhouettes. For these two strategies, only human body shape features can be extracted to address the insufficient input problem. Different from human silhouettes, skeleton models depict gaits from another different perspective. Human silhouettes can offer sufficient discriminative body shape information for gait recognition, while skeleton models enable to offer robust knowledge of human body structures. Using these two inputs can not only reduce the generalization error caused by insufficient input data, but also establish a scheme of mutual utilization and promotion between them. Advantages are retained, and disadvantages are restrained. Human silhouettes can be used to handle the sparsity problem for skeleton models, and skeleton models can be used to weaken the influence of appearance changes for human silhouettes. Thus, compared with the other two strategies, a more robust and comprehensive description can be acquired for each person through the combinations of segmented human silhouettes and extracted skeleton models.

From Table 1, it also can be seen that the averaged accuracy first monotonically arises as the input frame number increases. And the accuracy is approaching the best performance when the input samples consist of over 30 frames. For GaitSet, when 30 silhouettes are

sampled as its input, the accuracy of the NM, BG, and CL cases are 95.0%, 87.2%, and 70.4% separately. However, for the proposed method, when 10 silhouettes and 10 skeleton models are associated as its input, the accuracy of these three cases can respectively arise to 96.1%, 90.2%, and 71.2%. Although only a three times smaller number of original frames are utilized, our method still acquires a better performance than GaitSet in all cases. For GaitSet, 30 frames are required for 30 silhouettes. However, for the proposed method, 10 frames will suffice, because silhouettes and skeleton models can be integrated from the same input frames. The generalization error caused by insufficient input silhouettes can be suppressed by applying skeleton models. Also, silhouettes and skeleton models describe gaits from two different perspectives. A more robust and comprehensive gait description can be established for the same subject through their combinations. Therefore, although the input frame number is decreased, the proposed method still presents a prominent gait recognition performance. The other two strategies also obtain a better result than GaitSet. But since only human silhouettes are adopted, their improvement is not so prominent as the combinations of human silhouettes and skeleton models. According to these results, it can be concluded that compared with GaitSet, the proposed method has more potential to be explored into practical applications, since it can well approach the problem of insufficient input data.

Table 2 shows the results when different numbers of inputs are fed into the proposed network. Similar to above, when approaching the insufficient input problem, combining human silhouettes and skeleton models significantly outperforms the other two stategies, especially in the CL cases. Moreover, since the sum frame number is fixed to be 30, the averaged accuracy of horizontal flipping and CycleGAN appears much similar to GaitSet with 30 silhouettes solely being used. The reasons behind these conclusions are exactly the same as what we have claimed in the foregoing paragraphs. Human silhouettes can offer adequate discriminative body shape hints, and skeleton models can provide robust human body structure knowledge. Through their combinations, a full-scale portray can be established for each person, and the gait recognition performance can be improved although only insufficient input data can be acquired. For the other two strategies, features are extracted from human silhouettes only. Thus, similar to GaitSet, their performance tends to be stable as sufficient silhouettes are utilized. It offers us an alternative option of tackling this insufficient data problem, since skeleton extraction is not always robust.

In all, it can be concluded from these experiments that compared with GaitSet, the proposed method has more capability of addressing the problem of insufficient input data. Meanwhile, compared with generating more input silhouettes through horizontal flipping or CycleGAN, it is more robust and effective to assemble human silhouettes and skeleton modes as input when addressing this problem.

## Experiments on the OU-MVLP dataset

In this part, the proposed method is verified on the OU-MVLP Dataset when the same number of inputs are fed into the proposed network, i.e., $m = n$. Besides, based on the conclusions from the CASIA Gait Dataset B, for the aforementioned strategies, only the

**Table 2** Averaged rank-1 accuracies (%) on CASIA-B, excluding identical-view cases, using various numbers of provided gait frames.

| The number of segmented silhouettes ( $m$) | | 5 | 10 | 15 | 20 | 25 |
|---|---|---|---|---|---|---|
| The number of new generated frames ($n$) | | 25 | 20 | 15 | 10 | 5 |
| NM | Skeleton models | 95.7 | 96.6 | 96.6 | 96.5 | 96.6 |
| | Horizontal flipping | 95.4 | 95.3 | 95.4 | 95.4 | 95.1 |
| | CycleGAN | 95.5 | 95.4 | 95.6 | 95.3 | 95.0 |
| BG | Skeleton models | 88.2 | 90.2 | 90.4 | 90.4 | 90.8 |
| | Horizontal flipping | 88.3 | 88.7 | 88.7 | 89.0 | 88.2 |
| | CycleGAN | 89.4 | 89.4 | 89.0 | 88.6 | 88.5 |
| CL | Skeleton models | 68.6 | 71.8 | 74.6 | 75.6 | 74.5 |
| | Horizontal flipping | 69.0 | 69.3 | 69.2 | 68.8 | 69.4 |
| | CycleGAN | 68.9 | 69.0 | 69.6 | 68.2 | 71.7 |

strategy of combining human silhouettes and skeleton models as the network input will be evaluated in this part.

### Training and testing details

In these experiments, human silhouettes can be directly acquired from the original dataset. Skeleton models also can be directly established based on the extracted locations from *An et al. (2020)*. All input frames are also required to be aligned, cropped, and resized to $64 \times 44$ pixels, the same as the CASIA Gait Dataset B.

In the proposed network, for each branch, the channels of each convolution stage are set to 64, 128, and 256. For the global pipelines, the channels are correspondingly set to 128 and 256. The HPM scales are set to 5. For the training phase, each time a batch size of $8 \times 16$ sets is randomly sampled from our training set. Each sampled set contains $m$ human silhouettes and $n$ skeleton models. Adam is also used as our optimizer. The learning rate is $1e-4$ in the first 70K iterations, and changed into $1e-5$ in the last 30K iterations. The margin of $BA_+$ triplet loss is also set to 0.2. In the testing stage, all frames are utilized, and the batch size is correspondingly set to be 1.

### Results on the OU-MVLP dataset

Table 3 shows the comparison results of GaitSet and the proposed method when the same number of frames are utilized as the network input. It can be seen from this table that the proposed method highly outperforms GaitSet when only insufficient input data can be obtained. When 5 frames are utilized as input, the averaged accuracy of our proposed method is 78.0%, 8.2% higher than GaitSet. Also, when 10 frames are sampled as input, the averaged accuracy of our proposed method is 78.1%, 5.9% higher than GaitSet. For GaitSet, only human silhouettes are segmented from the input frames, thus the generalization error of each subject will be increased when only limited gait frames can be used. For the proposed method, however, human silhouettes and skeleton models can be extracted from the same input frames. Thus, although the input frame number is limited, it can also present a remarkable performance for gait recognition. Moreover, different from GaitSet only exploring human body shapes, a more comprehensive gait description is formed for

**Table 3** Averaged rank-1 accuracies (%) on OU-MVLP, excluding identical-view cases.

| The number of original input frames ($m$) | | GaitSet | | | | Proposed Method ($m = n$) | | | |
|---|---|---|---|---|---|---|---|---|---|
| | | 5 | 10 | 20 | 30 | 5 | 10 | 20 | 30 |
| | 0° | 49.9 | 53.9 | 56.4 | 56.3 | 66.4 | 66.0 | 66.3 | 66.8 |
| | 15° | 68.2 | 71.6 | 73.1 | 72.7 | 79.0 | 79.0 | 79.1 | 79.0 |
| | 30° | 76.9 | 79.0 | 79.8 | 79.4 | 82.9 | 83.2 | 83.2 | 83.1 |
| | 45° | 78.7 | 80.3 | 81.0 | 80.4 | 84.0 | 84.2 | 84.3 | 84.1 |
| | 60° | 70.9 | 73.5 | 74.4 | 73.9 | 80.4 | 80.5 | 80.5 | 80.7 |
| | 75° | 74.5 | 76.5 | 77.2 | 77.3 | 81.1 | 81.3 | 81.3 | 81.4 |
| Probe | 90° | 70.2 | 72.9 | 73.9 | 74.4 | 78.5 | 78.6 | 78.8 | 79.1 |
| | 180° | 53.2 | 57.2 | 60.1 | 60.5 | 66.4 | 65.8 | 66.3 | 66.9 |
| | 195° | 68.6 | 71.5 | 72.7 | 72.5 | 76.9 | 76.9 | 76.9 | 77.4 |
| | 210° | 76.0 | 77.8 | 78.7 | 78.2 | 79.2 | 79.5 | 79.6 | 79.7 |
| | 225° | 78.2 | 79.5 | 79.9 | 79.2 | 82.7 | 82.9 | 83.0 | 82.9 |
| | 240° | 70.2 | 72.5 | 73.3 | 72.8 | 78.9 | 79.1 | 79.2 | 79.2 |
| | 255° | 73.0 | 74.5 | 75.4 | 75.0 | 79.2 | 79.2 | 79.5 | 79.4 |
| | 270° | 68.0 | 70.5 | 71.7 | 71.7 | 76.4 | 76.4 | 76.6 | 76.9 |
| | AVE | 69.8 | 72.2 | 73.4 | 73.2 | 78.0 | 78.1 | 78.2 | 78.3 |

each subject in our proposed method. Human silhouettes can offer adequate inherent body shape hints for gait recognition, and skeleton models enable to offer robust knowledge of human body structures. Through their combinations, a full-scale portray of each person can be formed, and the gait recognition performance also can be improved. It also can be seen from this table that although only 5 frames are sampled, the averaged accuracy is already stabilized at its best performance, which illustrates that the proposed method is more robust than GaitSet on the OU-MVLP Dataset. In all, we can get the conclusion that compared with GaitSet, the proposed method is more capable of approaching the insufficient data problem for gait recognition.

## DISCUSSIONS

Relevant experiments have verified the robustness and effectiveness of our proposed method when handling the insufficient data problem. Compared with GaitSet, one of the world's best gait recognition methods, the proposed method also presents an excellent recognition performance when approaching this problem. Also, three different strategies are proposed in our method. Compared with the strategies of creating more human silhouettes using horizontal flipping or CycleGAN, the strategy of coupling human silhouettes and skeleton models as input is more robust and efficient when handling this insufficient data problem. However, for gait recognition, the problem of insufficient input data is still far from being handled. In real-world applications, human silhouettes and skeleton models are prominently influenced by segmentation and skeleton extraction algorithms. A unreliable algorithm will lead to a sharp degradation in the overall performance. Considering that lots of segmentation and skeleton extraction methods also offer the confidence scores when performing the segmentation and skeleton extraction tasks, a more feasible method

can be developed in the future work by combining human silhouettes and skeleton models according to their confidence scores. Moreover, some other methods also can be utilized to approach this insufficient data problem, e.g., image reconstruction, 3D human reconstruction, human parsing, etc.

## CONCLUSIONS

In the real world, due to many reasons, it is unlikely to always acquire sufficient gait frames for each person. Thus, for gait recognition, some measures are needed to enhance its performance when only a small number of gait frames are available. In this paper, a robust gait recognition method has been proposed to handle this insufficient data problem. First, in order to narrow the generalization error caused by insufficient input data, three different strategies are proposed in this paper to introduce more input frames. Moreover, a two-branch network is also proposed to formulate robust gait representations from the original and new generated input gait frames. Experiments on two of the most widely-used gait databases have illustrated that compared with GaitSet, one of the world's best gait recognition methods, the proposed method is more robust and efficient when tackling this insufficient data problem. Meanwhile, among the proposed three strategies, the strategy of assembling human silhouettes and skeleton models as input has been verified more robust and competent when approaching the problem of insufficient input data for gait recognition.

### Funding

This research project was supported by the Faculty of Information and Communication Technology, Mahidol University. There was no additional external funding received for this study. The funders had no role in study design, data collection and analysis, decision to publish, or preparation of the manuscript.

### Grant Disclosures

The following grant information was disclosed by the authors:
Faculty of Information and Communication Technology, Mahidol University.

### Competing Interests

The authors declare there are no competing interests.

### Author Contributions

- Lingxiang Yao conceived and designed the experiments, performed the experiments, analyzed the data, performed the computation work, prepared figures and/or tables, authored or reviewed drafts of the paper, and approved the final draft.
- Worapan Kusakunniran conceived and designed the experiments, performed the experiments, analyzed the data, prepared figures and/or tables, authored or reviewed drafts of the paper, and approved the final draft.

 

● Qiang Wu and Jian Zhang conceived and designed the experiments, performed the experiments, analyzed the data, authored or reviewed drafts of the paper, and approved the final draft.

## Data Availability

Code is available as a Supplemental File.

The data used in our experiments is available at the Center for Biometrics and Security Research: http://www.cbsr.ia.ac.cn/english/Gait%20Databases.asp.

## Supplemental Information

Supplemental information for this article can be found online at http://dx.doi.org/10.7717/peerj-cs.382#supplemental-information.

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
