# Peer review of "Gait recognition using a few gait frames"

_PeerJ Computer Science, doi:10.7717/peerj-cs.382_

## Round 0.1 · original submission · Major Revisions

The paper describes a method of gait recognition from a limited number of frames. Skeleton models are utilized in addition to silhouettes.

The new technical contribution is considered limited but the idea is found to be interesting. In general, it is a well-written paper. However, as noted by reviewer 2, the experimental design should be carefully handled. New experiments on other datasets and comparisons are required.

Please submit the paper again after a revision that addresses all the concerns of the reviewers.

·

Basic reporting

There is a difference between the title and the content of the text. The main content is described about the augmentation method, and the number of images is only expressed as a result. In the case that the point of claim is the number of images, the proposed network must have a structure that allows for time expansion and contraction.

For a limited number of gait frames, the author should refer to the following:
N. Akae, A. Mansur, Y. Makihara, Y. Yagi, ``Gait Recognition at One Frame per Second,'' In The 7th Int. Workshop on Robust Computer Vision (IWRCV2013), Jan. 2013.

Experimental design

It is quite possible that extracting skeleton features may be fail, but it is unclear how much failure occurred in the experiment.

Validity of the findings

no comment

Reviewer 2 ·

Basic reporting

This paper describes a method of gait recognition from a limited number of frames. For this purpose, the authors employ skeleton models in addition to silhouettes which are often used in gait recognition community as inputs for GaitSet network structure with horizontal pyramid mapping (HPM). Experimental results with CASIA-B show the effectiveness of the proposed method over horizontal flipping and CycleGAN.


Strength:
- Motivation of gait recognition from a limited number frames is well introduced and hence convincing.

Weaknesses:
- The authors misunderstand the notion of data augmentation.

- Important references are missing.

- Insufficient experimental validation

- The skeleton model is not well defined.

- Technical novelty is limited.


More details of the above-mentioned concern is as follows.


1. The authors misunderstand the notion of data augmentation.

The data augmentation usually refers to a technique to increase the diversity of your training set by applying random (but realistic) transformations such as image rotation. Hence, we cannot associate addition of a new branch with skeleton model with the data augmentation.

In addition, the reviewer is afraid that the general readers come up with an idea of increasing the number of input frames for gait recognition from a limited number of input frames regardless of training or test stage. This is because the problem setting makes the number of input frames limited and hence direct solution to this seems to be increase of the number of input frames, although this is different from the notion of data augmentation.

Therefore, the authors should clarify the contribution claims including but not limited to the above-mentioned points.


2. Important references are missing.

Because this study aims at solving gait recognition from a limited number of frames, the authors should cite more closely relevant work and also differentiate them from them.

For example, the following study tries reconstructing a gait energy image (GEI) from a limited number of frames and use the reconstructed GEI for recognition.

@article{Babaee_Neurocomp2019,
title = "Person identification from partial gait cycle using fully convolutional neural networks",
journal = "Neurocomputing",
volume = "338",
pages = "116 - 125",
year = "2019",
issn = "0925-2312",
doi = "https://doi.org/10.1016/j.neucom.2019.01.091",
url = "http://www.sciencedirect.com/science/article/pii/S0925231219301845",
author = "Maryam Babaee and Linwei Li and Gerhard Rigoll",
keywords = "Gait recognition, Gait energy image, Deep learning, Fully convolutional neural network",
}

In addition, the following study aims at the most extreme case, i.e., gait recognition from a single image by reconstructing a gait cycle.

C. Xu, Y. Makihara, X. Li, Y. Yagi, J. Lu, ``Gait Recognition from a Single Image using a Phase-Aware Gait Cycle Reconstruction Network,'' Prof. of the 16th European Conf. on Computer Vision (ECCV 2020), Glasgow, UK (online), Aug. 2020.


Moreover, the following family of gait recognition from low frame-rate video is also closely relevant with this work, since they also handle a limited number of input frames.

@inproceedings{Akae_CVPR2012,
author = {N. Akae and A. Mansur and Y. Makihara and Y. Yagi},
title = {Video from Nearly Still: an Application to Low Frame-rate Gait Recognition},
booktitle = {Proc. of the 25th IEEE Conf. on Computer Vision and Pattern Recognition (CVPR 2012)},
year = {2012},
pages = {1537-1543},
address = {Providence, RI, USA},
month = {Jun.}
}

@INPROCEEDINGS{Guan_IWBF2013,
author={Yu Guan and Chang-Tsun Li and Choudhury, S.D.},
booktitle={Biometrics and Forensics (IWBF), 2013 International Workshop on},
title={Robust gait recognition from extremely low frame-rate videos},
year={2013},
month={April},
pages={1-4},
doi={10.1109/IWBF.2013.6547319},
}


- Insufficient experimental validation

The authors made experimental validation only with a single gait database, i.e., CASIA-B. It would be better to do more comprehensive experimental validation., since there are many other larger-scale gait databases including but not limited to the followings.

@article{Hofmann_JVCIR2014,
author = {Hofmann, Martin and Geiger, J\"{u}rgen and Bachmann, Sebastian and Schuller, Bj\"{o}rn and Rigoll, Gerhard},
title = {The TUM Gait from Audio, Image and Depth (GAID) Database: Multimodal Recognition of Subjects and Traits},
journal = {J. Vis. Comun. Image Represent.},
issue_date = {January, 2014},
volume = {25},
number = {1},
month = jan,
year = {2014},
issn = {1047-3203},
pages = {195--206},
numpages = {12},
url = {http://dx.doi.org/10.1016/j.jvcir.2013.02.006},
doi = {10.1016/j.jvcir.2013.02.006},
acmid = {2565863},
publisher = {Academic Press, Inc.},
address = {Orlando, FL, USA},
keywords = {Acoustic gait recognition, Depth gradient histogram energy image, Gait energy image, Gait recognition, Multimodal fusion, Soft biometrics},
}

@Article{Takemura_CVA2018,
author="Takemura, Noriko
and Makihara, Yasushi
and Muramatsu, Daigo
and Echigo, Tomio
and Yagi, Yasushi",
title="Multi-view large population gait dataset and its performance evaluation for cross-view gait recognition",
journal="IPSJ Transactions on Computer Vision and Applications",
year="2018",
month="Feb",
day="20",
volume="10",
number="1",
pages="4",
abstract="This paper describes the world's largest gait database with wide view variation, the ``OU-ISIR gait database, multi-view large population dataset (OU-MVLP)'', and its application to a statistically reliable performance evaluation of vision-based cross-view gait recognition. Specifically, we construct a gait dataset that includes 10,307 subjects (5114 males and 5193 females) from 14 view angles ranging 0{\textdegree} -90{\textdegree}, 180{\textdegree} -270{\textdegree}.",
issn="1882-6695",
doi="10.1186/s41074-018-0039-6",
url="https://doi.org/10.1186/s41074-018-0039-6"
}

@Article{Xu_CVA2017,
author="Xu, Chi
and Makihara, Yasushi
and Ogi, Gakuto
and Li, Xiang
and Yagi, Yasushi
and Lu, Jianfeng",
title="The OU-ISIR Gait Database comprising the Large Population Dataset with Age and performance evaluation of age estimation",
journal="IPSJ Transactions on Computer Vision and Applications",
year="2017",
month="Dec",
day="21",
volume="9",
number="1",
pages="24",
abstract="In this paper, we describe the world's largest gait database, the ``OU-ISIR Gait Database, Large Population Dataset with Age (OULP-Age)'' and its application to a statistically reliable performance evaluation of gait-based age estimation. Whereas existing gait databases include only 4016 subjects at most, we constructed an extremely large-scale gait database that includes 63,846 subjects (31,093 males and 32,753 females) with ages ranging from 2 to 90 years old. Benchmark algorithms of gait-based age estimation were then implemented to evaluate statistically significant performance differences. Additionally, the dependence of gait-based age estimation performance on gender and age group, in addition to the number of training subjects, was investigated to provide several insights for future research on the topic.",
issn="1882-6695",
doi="10.1186/s41074-017-0035-2",
url="https://doi.org/10.1186/s41074-017-0035-2"
}

@Article{Uddin_CVA2018,
author="Uddin, Md. Zasim
and Ngo, Thanh Trung
and Makihara, Yasushi
and Takemura, Noriko
and Li, Xiang
and Muramatsu, Daigo
and Yagi, Yasushi",
title="The OU-ISIR Large Population Gait Database with real-life carried object and its performance evaluation",
journal="IPSJ Transactions on Computer Vision and Applications",
year="2018",
month="May",
day="30",
volume="10",
number="1",
pages="5",
issn="1882-6695",
doi="10.1186/s41074-018-0041-z",
url="https://doi.org/10.1186/s41074-018-0041-z"
}

3. Technical soundness of use of the skeleton model under the problem setting is unclear.

The authors introduce the skeleton model as a countermeasure for a limited number of input frames. The reviewer understands that inclusion of the skeleton model as inputs may increase representation capability and hence improves gait recognition accuracy. On the other hand, the reviewer does not find clear connection between the inclusion of the skeleton model and a limited number of frames. For example, the reviewer imagines that the inclusion of the skeleton model is just generally beneficial for various settings, e.g., clothing invariant gait recognition with a full gait cycle. So, the proposed method does not seem to be a specific countermeasure for this problem setting, and hence this reduces the technical soundness of this method.


4. The skeleton model is not well defined.

The authors just illustrate the skeleton model in Fig. 2 without detailed explanations. The reviewer actually does not understand what the skeleton model indicates and how the skeleton model is created. Since the addition of the skeleton model seems to be a key component of this paper, the not well-defined skeleton model is critical.


5. Technical novelty is limited.

The proposed network architecture is an extension of GaitSet for two-stream networks of silhouette and the skeleton model. The GaitSet and two-stream networks themselves are existing and well known modules, and the way of combination is also straightforward, and hence the technical novelty of this paper seems to be limited.


6. Minor editorial errors and suggestions for improvement.

- l. 32: Work (We et al., 2016) is irrelevant with criminal investigation. More suitable citation could be the followings.

@article{Bouchrika_FS2011,
author = {Bouchrika, I. and Goffredo, M. and Carter, J. and Nixon, M.},
title = {On Using Gait in Forensic Biometrics},
journal = {Journal of Forensic Sciences},
volume = {56},
number = {4},
pages = {882-889},
year = {2011},
}

@inproceedings{Larsen_SPIE2007,
author = {Peter K. Larsen and Erik B. Simonsen and Niels Lynnerup},
title = {{Gait analysis in forensic medicine}},
volume = {6491},
booktitle = {Videometrics IX},
editor = {J.-Angelo Beraldin and Fabio Remondino and Mark R. Shortis},
organization = {International Society for Optics and Photonics},
publisher = {SPIE},
pages = {195 -- 202},
keywords = {Gait analysis, gait recognition, forensic medicine, biometric, recognition, joint angles, posture},
year = {2007},
doi = {10.1117/12.698512},
URL = {https://doi.org/10.1117/12.698512}
}

- Figure 1: This seems to be reuse from GaitSet paper. If so, the authors need to get reuse permission from publisher (e.g., through RightsLink).

- A term human appearance would be better to be replaced with a body shape, since appearance may be associated not only with the body shape but also with cloths color and textures.

- l. 61: learning based -> learning-based

- l. 78: all 11 probe views. This seems to refer to CASIA-B database. If so, it would be better to clarify it.

- figure -> Figure or Fig.

Experimental design

See the above basic reporting.

Validity of the findings

See the above basic reporting.

Additional comments

See the above basic reporting.

·

Basic reporting

This paper proposes a robust method to improve the performance of gait recognition when only a small number of frames can be offered. Two most widely used image augmentation methods and combining people’s silhouettes and skeleton models as input are compared in this paper. It is an important problem for gait recognition since in the real world it is impossible to acquire sufficient gait frames for each person due to various factors. This paper is well organised. Literature references and sufficient field background/context are provided. Experiment results are convincing, and the conclusions are well stated. Throughout the whole paper, professional English is well used.

Experimental design

The research problem in this paper is well defined, relevant and meaningful. It is innovative because few methods have been proposed for gait recognition to investigate the problem of insufficient input data. Methods are described with sufficient details. The experiments are well designed, and the experiment results provided in this paper are convincing. Moreover, the authors provide the implementation codes that make the proposed methods reproducible.

Validity of the findings

This paper is innovative, since for gait recognition few methods have been proposed to investigate the problem of insufficient input data. Two most widely used image augmentation methods and combining people’s silhouettes and skeleton models as input are compared in this paper. One conclusion has been made that using human silhouettes and their skeleton models as the network input is the most effective way to improve the gait recognition performance when the input frame number is limited. Literature references and sufficient field background/context are clearly stated. All underlying data have been provided. They are robust, statistically sound, and well controlled. The experiment results support the effectiveness of the proposed method. Conclusions are well stated and connected with the proposed research problem.

Additional comments

There are some minor errors and typos, for example,
1) F_1, F_2, …, and F_3^’ in Line 177 should be marked on Fig.3.
2) In Line 203, it should be Section 1.1.

---

## Round 0.2 · Minor Revisions

Dear authors,

Please include further explanations in your revised version to address the concerns of reviewer 1.

Reviewer 2 ·

Basic reporting

Although the manuscript is updated and get a better shape, the following concerns remain unsolved.

3. Technical soundness of use of the skeleton model under the problem setting is unclear.

Explanation provided in the response letter and the revised manuscript is still unconvincing for a countermeasure for a limited number of intput frames. The authors seem to explain the performance drops for a limited number of input frames and improve the performance by skeleton information. But the inclusion of skeleton information is not relevant with the shortage of input frames. So, it is still strange to me to include skeleton information in the context of gait recognition from a limited number of frames.


4. The skeleton model is not well defined.

The motivation of the skeleton model is described in the revised manuscript, but its actual contents are still not described (e.g., the number of joints, representation, etc.).


5. Technical novelty is limited.

Although the authors list up the contributions in the introduction section, the technical novelty concerns raised up in the first-round review are not addressed at all.

Experimental design

N/A

Validity of the findings

N/A

Additional comments

N/A

·

Basic reporting

This paper proposes a robust method to improve the performance of gait recognition when only a small number of frames are available. It is a significant problem for gait recognition because in the real world it is impossible to obtain sufficient gait frames for each person due to various factors. Three strategies are proposed in this paper to approach this problem, including horizontally flipping silhouettes, generating more silhouettes using CycleGAN, and combining human silhouettes and skeleton models. This paper is well organized, and professional English is well. Literature references and field background/context are offered. Experiment results are convincing, and the conclusions are well stated.

Experimental design

In this paper, the research problem in this paper is well defined. It is innovative since few methods have been proposed for gait recognition to investigate the problem of insufficient input data. In this paper, methods are explained with sufficient details, and the experiments are well designed. The experiment results provided in this paper are convincing. Moreover, the authors provide the implementation codes that make the proposed methods reproducible.

Validity of the findings

This paper is innovative, since for gait recognition few methods have been proposed to handle the problem of insufficient input data. Three strategies are proposed in this paper, including horizontally flipping silhouettes, generating more silhouettes using GAN-based networks, and combining human silhouettes and skeleton models as the network input. One conclusion is made that utilizing human silhouettes and skeleton models as input is the most efficient way to enhance the gait recognition performance when the input frame number is limited. Literature references and field backgrounds are clearly stated. All provided data are statistically sound, and well controlled. The experiment results support the effectiveness of the proposed method. Conclusions are well stated and related with the proposed research problem.

Additional comments

The authors have eased my questions. The paper can be accepted now.

---

## Round 0.3 · accepted · Accept

Your paper is accepted to be published.